# Evaluation of the Accuracy and Reliability of Responses Generated by Artificial Intelligence Related to Clinical Pharmacology

**DOI:** 10.3390/jcm14217563

**Published:** 2025-10-25

**Authors:** Michal Ordak, Julia Adamczyk, Agata Oskroba, Michal Majewski, Tadeusz Nasierowski

**Affiliations:** 1Department of Pharmacotherapy and Pharmaceutical Care, Faculty of Pharmacy, Medical University of Warsaw, Banacha 1 Str., 02-097 Warsaw, Poland; s083129@student.wum.edu.pl (J.A.); agata.oskroba@wum.edu.pl (A.O.); s082570@student.wum.edu.pl (M.M.); 2Department of Psychiatry, Faculty of Pharmacy, Medical University of Warsaw, Nowowiejska 27 Str., 00-665 Warsaw, Poland; tadeusz.nasierowski@wum.edu.pl

**Keywords:** artificial intelligence, clinical pharmacology, drug interactions

## Abstract

**Background/Objectives:** Artificial intelligence (AI) is gaining importance in clinical pharmacology, supporting therapeutic decisions and the prediction of drug interactions, although its applications have significant limitations. The aim of the study was to evaluate the accuracy of the responses of four large language models (LLMs), namely ChatGPT-4o, ChatGPT-3.5, Gemini Advanced 2.0, and DeepSeek, in the field of clinical pharmacology and drug interactions, as well as to analyze the impact of prompting and questions from the National Specialization Examination for Pharmacists (PESF) on the results. **Methods:** In the analysis, three datasets were used: 20 case reports of successful pharmacotherapy, 20 reports of drug–drug interactions, and 240 test questions from the PESF (spring 2018 and autumn 2019 sessions). The responses generated by the models were compared with source data and the official examination key and were independently evaluated by clinical-pharmacotherapy experts. Additionally, the impact of prompting techniques was analyzed by expanding the content of the queries with detailed clinical and organizational elements to assess their influence on the accuracy of the obtained recommendations. **Results:** The analysis revealed differences in the accuracy of responses between the examined AI tools (*p* < 0.001), with ChatGPT-4o achieving the highest effectiveness and Gemini Advanced 2.0 the lowest. Responses generated by Gemini were more often imprecise and less consistent, which was reflected in their significantly lower level of substantive accuracy (*p* < 0.001). The analysis of more precisely formulated questions demonstrated a significant main effect of the AI tool (*p* < 0.001), with Gemini Advanced 2.0 performing significantly worse than all other models (*p* < 0.001). An additional analysis comparing responses to simple and extended questions, which incorporated additional clinical factors and the mode of source presentation, did not reveal significant differences either between AI tools or within individual models (*p* = 0.34). In the area of drug interactions, it was also shown that ChatGPT-4o achieved a higher level of response accuracy compared with the other tools (*p* < 0.001). Regarding the PESF exam questions, all models achieved similar results, ranging between 83 and 86% correct answers, and the differences between them were not statistically significant (*p* = 0.67). **Conclusions**: AI models demonstrate potential in the analysis of clinical pharmacology; however, their limitations require further refinement and cautious application in practice.

## 1. Introduction

Artificial intelligence (AI) has created new perspectives in medicine, introducing revolutionary changes in diagnostics and treatment. Consequently, healthcare workers can cope more effectively with problems encountered in hospitals, such as disease management, service accessibility, improvement of treatment methods, and cost reduction. Advanced AI technologies, such as machine learning and deep learning, have played a key role in diagnostics, patient monitoring, drug discovery, and public health support systems. AI has brought significant progress in clinical decision support systems and disease prediction models, helping to identify conditions and directing early interventions [1,2,3]. In recent years, an increasing number of articles on the role of AI in clinical pharmacy have been published [4]. Studies indicate that artificial intelligence and machine learning have great potential in the area of precise drug dosing and therapeutic drug monitoring. However, the results generated by algorithms are still difficult to interpret unambiguously and do not always allow accurate prediction of concentrations. It is emphasized that further development of interoperability and data exchange between medical institutions will be crucial for the effective implementation of these technologies in clinical practice. [5]. Neural networks are capable of generating concentration–time curve predictions without the need to use classical pharmacokinetic models. Reports also indicate that AI tools can operate effectively on small datasets and predict new dosing regimens, paving the way for faster analyses and practical applications in therapy [6]. The results of some analyses suggest that ChatGPT can effectively identify drug interactions, provide personalized therapeutic recommendations, and support patient treatment monitoring and management. Importantly, the model correctly solved all analyzed clinical cases, both simple and complex, demonstrating high effectiveness in providing accurate answers [7]. However, various observations indicate that ChatGPT does not cope well with the practical problems faced by clinical pharmacists, and the responses it provides are often incomplete or incorrect. Low repeatability of generated content and an accuracy level insufficient to ensure safe clinical care are also emphasized [8]. Another study found that the average rate of correct ChatGPT responses to clinical pharmacy questions was only 44.9%, confirming its limited usefulness in this field. Meanwhile, an analysis of real clinical cases showed that the tool performed relatively well in drug-related counseling but considerably worse in prescription assessment, adverse reaction recognition, and patient education [9]. The question of whether language models provide reliable information about drugs therefore remains uncertain. ChatGPT, in its current form, does not ensure sufficient accuracy and completeness of responses, which significantly limits its practical application [10]. On the other hand, newer generations, such as GPT-4o, indicate the possibility of more effective use of AI for drug identification and providing therapeutic recommendations to patients [11]. In recent years, new large language models such as Gemini Advanced 2.0 and DeepSeek have emerged as potential alternatives to ChatGPT. However, studies directly comparing these tools in terms of response accuracy in clinical pharmacotherapy are still lacking.

The main aim of this study was to evaluate how effectively these artificial intelligence systems can propose appropriate pharmacotherapy based on the content of scientific articles.

The study also examined three additional aspects: the impact of detailed query formulation (prompting) on the accuracy of responses, the ability of the models to identify drug interactions from scientific publications, and their performance in answering questions from the State Specialization Examination for Pharmacists in the field of hospital pharmacy.

Comparing the results obtained for different models allowed us to identify their strengths and limitations in the context of clinical pharmacy practice.

## 2. Materials and Methods

### 2.1. Models and Datasets Used

The following AI models and versions were used in the study: ChatGPT-3.5 (OpenAI, San Francisco, CA, USA), ChatGPT-4o (May 2025 release; OpenAI, San Francisco, CA, USA), Gemini Advanced 2.0 (Google DeepMind, London, UK), and DeepSeek V2.5 (DeepSeek Technologies, Hangzhou, Zhejiang, China). All models were accessed and tested in May 2025 using their publicly available web-based versions current at that time. Three datasets were analyzed:-A set of 20 descriptions with documented patient cases that were successfully treated with pharmacotherapy [12,13,14,15,16,17,18,19,20,21,22,23,24,25,26,27,28,29,30,31]. The articles were selected in such a way as to include an unambiguous description of the applied treatment that led to the patient’s recovery. The following formula was used to search scientific articles on PubMed: (“case reports” [Publication Type] OR “case study” [Title/Abstract]) AND (“pharmacotherapy” [MeSH Terms] OR “drug therapy” [MeSH Terms] OR “pharmacological treatment” [Title/Abstract]) AND (“successful treatment” [Title/Abstract] OR “recovery” [Title/Abstract] OR “cured” [Title/Abstract] OR “complete remission” [Title/Abstract]). The time range of the analyzed articles covers the years 2018–2025. The query intended for the AI models was accompanied by a description of the patient’s medical history taken from the publication.-A set of 20 descriptions containing clinical cases of patients in whom drug–drug interactions were diagnosed [32,33,34,35,36,37,38,39,40,41,42,43,44,45,46,47,48,49,50,51]. In each case, the cause was diagnosed and an effective therapeutic intervention leading to recovery was applied. The following formula was used to search scientific articles on PubMed: “drug interaction” AND “case report” AND (resolved OR treated OR successful). At the same time, the time range of the analyzed articles was set to the years 2018–2025. The query was accompanied by a description of the patient’s medical history.-A set of PESF test questions, hospital pharmacy, spring session 2018 and autumn session 2019, each containing 120 questions [52].

All AI models received identical prompts formulated in English, copied verbatim to each platform to ensure full consistency of input and comparability of responses. These datasets were selected to enable a comprehensive assessment of the models’ ability to provide accurate and clinically relevant responses in various pharmacotherapy contexts, including treatment selection, identification of drug interactions, and examination-based knowledge.

### 2.2. Study of Pharmacotherapy Accuracy

This study evaluated the accuracy of AI-generated answers for therapy selection, using selected scientific publications as a reference. The obtained answers were compared with the medical interventions that had been undertaken in patients. AI tools scored highest when their proposed therapies matched those described in the literature, and lowest when they recommended procedures inconsistent with pharmacological guidelines. Intermediate scores were awarded to AI models in situations where they proposed management consistent with current treatment standards, but different from that indicated by the authors of the analyzed publication as optimal for the given case, since the assessment focused on the ability to reproduce the specific therapeutic approach described in the reference case rather than general adherence to clinical guidelines. The verification of answers and the assignment of scores were carried out by independent authors of the publication, all experienced specialists in clinical pharmacotherapy working within the same academic unit. A 0–10 scoring scale was used to reflect the degree of concordance between AI-generated recommendations and those described in the reference publications, where 0 indicated a completely incorrect or clinically unsafe response, 10 indicated full agreement with the therapeutic approach reported in the source article, and intermediate values represented partial correctness or incomplete clinical reasoning. In cases of uncertainty or divergent interpretations, the evaluators met to conduct an expert discussion and jointly established the final score by consensus. In the study, five questions were developed, each serving a separate methodological function. The first question aimed to assess the extent to which artificial intelligence tools are able to indicate the correct pharmacotherapy for a specific disease entity, regardless of the context of the individual patient. The second question was intended to verify the ability of the models to predict the appropriate treatment regimen based on a detailed clinical history, which made it possible to determine whether the tools actually adjust their recommendations to a specific case or limit themselves to presenting general information. The third question focused on assessing the correctness of pharmacotherapy dosing and also covered practical aspects of drug use, such as the administration schedule, the need for additional tests, and vaccinations. The fourth question analyzed the models’ ability to carry out the reasoning process leading to the appropriate choice of therapy, as well as to correctly explain the pathophysiological mechanisms of the disease and the action of drugs. Finally, the fifth question, on the other hand, concerned the sources of knowledge used by AI tools when formulating therapeutic recommendations, considering their reliability and quality, with the most valuable being considered official guidelines, recommendations of scientific societies, and peer-reviewed publications. Together, these questions provided a multidimensional evaluation of the models’ ability to generate clinically relevant and accurate therapeutic recommendations, directly addressing the main aim of the study.

### 2.3. Evaluation of the Importance of Prompting in Generating Responses by AI Models

In questions 6–9, the phenomenon of prompting was analyzed, defined as the process of formulating effective queries addressed to language models, which enables artificial intelligence-based systems to generate responses that better match the user’s intentions [53,54]. The construction of a prompt may include not only the content of the question but also a description of the clinical situation or organizational context, which facilitates the model’s deeper understanding of the problem and promotes obtaining a response adequate to the presented scenario. In this part of the study, the same articles as in questions 1–5 were used, allowing for a direct comparison of the results. Question 6 expanded on question 2 by adding contextual elements such as the country in which the therapy was conducted, the hospital’s budget and related limitations in access to drugs or equipment, the availability of an appropriate medical team, and the patient’s willingness to participate in experimental treatment methods. This design made it possible to examine to what extent the precision and detail of formulating a question affects the substantive quality of the model’s response. Questions 7 and 8 were designed to assess whether the prompting techniques applied in question 6 significantly influenced the content and quality of the recommendations, and the results of these analyses could be compared with the outcomes concerning questions 3 and 4. Question 9 evaluated whether the sources of information indicated by the models could be presented more usefully and accessibly, with emphasis on ensuring active links, free access to full texts, and eliminating subscription barriers. Its construction enabled a direct comparison with question 5, which allowed assessing whether increasing the detail of the instructions improved the quality of the models’ responses.

For full transparency and reproducibility, the exact prompts used in this part of the study were as follows:-*Question 6*: “Imagine you are a physician. A patient presents at your hospital. After reviewing his medical records, propose a patient-specific treatment plan, keeping in mind our hospital’s budget constraints and lack of cutting-edge technology. The following photo illustrates the patient’s appearance. Suggest the optimal solution.”-*Question 7*: “What therapy can be proposed for the patient, considering their medical history as presented in the following text? What is the recommended medication dosage and duration?”-*Question 8*: “Explain the reasoning behind your choice of the most appropriate therapy for the patient. Why is this therapy the best option?”-*Question 9*: “Provide the source of the information and assure me that I can easily find it on the internet.”

All prompts were formulated in English and entered identically into each AI model to ensure consistency and comparability of responses.

To illustrate this approach, question 3 was selected, which aimed to assess the ability of AI tools to correctly determine the dosing of pharmacotherapy in the context of an individual clinical case. The content of the question also included a request to prepare notes intended both for the patient and medical staff, which was aimed at prompting the model to provide practical recommendations, such as drug administration times, the need for additional tests, or prior vaccinations. When AI tools in earlier stages failed to propose appropriate therapy based on the source article, they were asked to indicate the correct dosing of a specific pharmacotherapy, taking into account the patient’s characteristics. An example of an incorrect response in this respect was the recommendation provided by the ChatGPT 3.5 model, presented in the figure below (Figure 1). In the analyzed case, the publication described a patient with SAPHO syndrome successfully treated with a combination of intramuscular betamethasone (7 mg, 1 mL) during the acute phase, followed by oral abrocitinib 100 mg once daily for several months and minocycline 50 mg twice daily for 8 weeks [29]. ChatGPT 3.5, however, proposed a loading dose of abrocitinib 200 mg, recommended oral rather than injectable betamethasone for only two weeks, and failed to specify the total duration of treatment. While the model correctly identified the dosing of minocycline and suggested appropriate follow-up intervals, its overall therapeutic proposal did not align with the clinically validated treatment described by the physicians in the original case. This example illustrates how AI-generated pharmacotherapy recommendations can diverge from real-world clinical management, highlighting the need for expert supervision in their interpretation. This example illustrates how the detailed formulation of prompts can influence the accuracy and clinical adequacy of AI-generated pharmacotherapy recommendations, highlighting the central role of prompting in improving model performance.

### 2.4. Study of Drug Interactions

In the next part of the study, the focus was on assessing the ability of AI models to identify and interpret drug–drug interactions and on analyzing the quality of the sources used in generating responses. The first question tested whether the tools could independently recognize potential interactions based on clinical symptoms and the list of medications used by the patient, without directly indicating their presence in the task description. The second question aimed to verify whether the models were able to appropriately adjust treatment after identifying an interaction by discontinuing the drug responsible for its occurrence, reducing its dose, or, in justified cases, suggesting a safe alternative. In situations leading to toxic effects, the tools were additionally expected to propose an appropriate antidote or cleansing procedures aimed at minimizing the risk of complications. The third question concerned the ability of the models to correctly explain the mechanisms underlying drug interactions. The assessment was based on comparing the generated explanations with the mechanisms described in source publications and in the authors’ clinical commentary. The fourth question assessed the reliability and timeliness of the sources cited by the models. This included verifying whether references were accurate and active, which allowed for the identification of potentially fictitious or outdated sources. AI models achieved the highest scores in cases where they correctly recognized the presence of interactions and clearly identified all the drugs responsible for them, in accordance with the analyzed articles. The lowest scores were assigned to AI tools that did not identify the interactions or presented incorrect information.

To illustrate the method of analyzing drug interactions, an example presented in Figure 2 was used. The original publication described an 8-year-old male patient diagnosed with community-acquired pneumonia and asthma exacerbation, who was treated with a prednisolone elixir (containing 5% ethanol *v*/*v*) followed approximately 2.5 h later by an intravenous infusion of ceftriaxone. Shortly after the ceftriaxone administration, the patient developed facial flushing that resolved with diphenhydramine, initially misdiagnosed as an allergic reaction to the antibiotic. Upon further evaluation, the event was correctly identified as a mild disulfiram-like reaction. This response was generated by the ChatGPT 3.5 model, which did not recognize the existing drug interaction. In the analyzed case, the correct identification was a disulfiram-like reaction, caused by the interaction of an oral prednisolone elixir containing 5% ethanol *v/v* and ceftriaxone administered intravenously approximately 2.5 h later. ChatGPT 3.5 misinterpreted the patient’s symptoms, suggesting generalized staphylococcal pneumonia as the main cause. The model pointed to a reaction to ceftriaxone only secondarily, classifying it as an allergic reaction rather than as a result of interaction with ethanol present in prednisolone. In contrast, the clinical reassessment of the case revealed that the symptoms represented a mild disulfiram-like reaction, with facial flushing resolving after diphenhydramine administration and no recurrence following the discontinuation of ethanol-containing medication. The antibiotic therapy was subsequently changed to azithromycin, with no further complications observed. The response was scored 3/10, as the tool did not recognize the key drug interaction. Importantly, the model’s interpretation resembled the incorrect diagnosis previously made by doctors in the first clinic, who also failed to establish the correct cause of the symptoms [38]. This case illustrates how AI models may replicate diagnostic errors observed in real-world clinical settings and emphasizes the necessity of expert oversight in interpreting AI-generated pharmacotherapy recommendations.

### 2.5. Study of the Accuracy of Answers to the Hospital Pharmacy Examination

The study aimed to evaluate the effectiveness of AI models in solving examination questions for hospital pharmacists. The analysis was based on questions from the PESF in the field of hospital pharmacy specialization [52]. Each question was entered individually into the selected AI models. Answers were considered correct if they matched the official PESF answer key.

### 2.6. Statistical Analysis

The statistical analysis was performed using the IBM SPSS Statistics 25 package. To verify whether there was a statistically significant relationship between the categorical levels of substantive correctness of answers (high, intermediate, low) and the type of AI tool, the chi-square test was applied. Two-way analysis of variance made it possible to assess whether AI tools and their interaction with the type of analyzed questions had a statistically significant effect on the correctness of the answers provided. When a statistically significant interaction was observed, simple main effects analysis was conducted for further examination. Pairwise comparisons within the simple main effects analysis were adjusted using the Bonferroni correction to control for Type I error. This allowed assessment whether the effect of one factor on the dependent variable depended on the level of the other factor. This approach enabled the identification of complex relationships between factors that are not revealed by main effects analyzed separately. A value of *p* < 0.05 was adopted as the level of statistical significance.

## 3. Results

### 3.1. Correctness of Pharmacotherapy

In order to assess whether the type of question and the AI tool have a statistically significant impact on the correctness of the provided answers, a two-way analysis of variance was applied. The conducted analysis revealed the occurrence of a statistically significant main effect of the AI tool, F(3;376) = 16.68; *p* < 0.001. Pairwise comparisons showed that the mean number of correct answers provided by Gemini Advanced 2.0 was statistically significantly lower in relation to each of the other tools (*p* < 0.001). As shown in Figure 3, ChatGPT-4o achieved the highest median number of correct answers, while Gemini Advanced 2.0 achieved the lowest.

A statistically significant relationship was observed between the level of substantive correctness of answers and the AI tool, χ^2^(6) = 35.17; *p* < 0.001. The highest level was observed for ChatGPT-4o, while the lowest concerned ChatGPT 3.5 and DeepSeek (Figure 4). Notably, for Gemini Advanced 2.0, most answers showed a low level of substantive correctness, χ^2^(2) = 49.88; *p* < 0.001. In the “low” category, the distribution of responses across the tools was as follows: ChatGPT-4o—13%; ChatGPT-3.5—7%; DeepSeek—17%; Gemini Advanced 2.0—63%. For the “intermediate” level, the proportions were 23%, 26%, 25%, and 26%, respectively, while for the “high” level, they were 33%, 28%, 26%, and 13%. In the *low* category, the distribution of responses across the tools was as follows: ChatGPT-4o—13%; ChatGPT-3.5—7%; DeepSeek—17%; Gemini Advanced 2.0—63%. For the *moderate* level, the proportions were 23%, 26%, 25%, and 26%, respectively, while for the *high* level, they were 33%, 28%, 26%, and 13%. Percentages presented in Figure 4 are normalized within each accuracy category (low, intermediate, high), with the values across tools summing to 100% for each category.

No statistically significant interaction was observed between the AI tool and the type of analyzed question, F(12;376) = 1.1; *p* = 0.36. Thus, there were no statistically significant differences between individual AI tools in the scope of each of the five questions separately. Similarly, no differences were observed regarding the correctness of the provided answers for each AI tool separately between the individual questions. The mean number of correct answers remained comparable for each question between AI tools, as well as within each AI tool separately across the five questions.

### 3.2. Detailed Questions Concerning Pharmacotherapy

The next analysis focused on more precisely formulated questions. A statistically significant main effect of the AI tool was also observed, F(3;304) = 19.78; *p* < 0.001; eta^2^ = 0.16 (Table 1). Pairwise comparisons showed that the mean of correct answers provided by Gemini Advanced 2.0 was statistically significantly lower in relation to each of the other tools separately (*p* < 0.001). A similar effect was found for DeepSeek compared with ChatGPT 3.5 (*p* = 0.03).

These results were confirmed by a statistically significant relationship between the level of substantive correctness of answers and the AI tool, χ^2^(6) = 35.86; *p* < 0.001. Based on the chart below, the lowest level of substantive correctness of answers concerns the Gemini Advanced 2.0 tool. For this tool, most results indicated a low level of substantive correctness of answers, χ^2^(2) = 12.48; *p* = 0.002 (Figure 5). In the *low* category, the distribution of responses across the tools was as follows: ChatGPT-4o—8%; ChatGPT-3.5—8%; DeepSeek—23%; Gemini Advanced 2.0—62%. For the *moderate* level, the proportions were 25%, 29%, 25%, and 31%, respectively, while for the *high* level, they were 28%, 33%, 25%, and 14%. Percentages presented in Figure 5 are normalized within each accuracy category (low, moderate, high), with the values across tools summing to 100% for each category.

The further conducted analysis did not show a statistically significant interaction of the AI tool with the type of question asked, F(9;304) = 1.27; *p* = 0.25. There are no differences between individual AI tools for each of the four questions separately. Likewise, the comparison of the effectiveness of answers within a single AI tool between the questions did not show significant differences. The mean number of correct answers remains at a similar level both between AI tools for each question and within each AI tool for all five questions.

In the present study, an additional analysis consisting of comparing question 2, assessing the ability of the models to predict an appropriate treatment regimen based on clinical history, with question 6, which also considered systemic and individual factors, was carried out. Similarly, question 5, concerning the reliability and quality of knowledge sources, was compared with question 9, which examined how these sources were presented in terms of accessibility and usefulness. The conducted analysis of variance did not show the occurrence of a statistically significant interaction between the AI tool and these four distinguished questions, F(9;304) = 1.13; *p* = 0.34. Thus, no statistically significant differences were observed between the indicated questions within individual AI tools. The obtained mean results of answers to questions 5 and 9 as well as 2 and 6 remain similar in the case of individual AI tools.

The first step assessed whether individual AI tools differ from each other in terms of the correctness of the answers provided to the analyzed questions. The conducted two-way analysis of variance revealed a statistically significant main effect of the AI tool, F(3;304) = 42.87; *p* < 0.001; eta^2^ = 0.18, as well as a statistically significant interaction between the analyzed factors, F(9;304) = 8.39; *p* < 0.001; eta^2^ = 0.2. The pairwise comparisons showed that the ChatGPT-4o tool obtained a higher mean (Figure 6) of correct answers in relation to ChatGPT 3.5 (*p* < 0.001), Gemini Advanced 2.0 (*p* < 0.001), and DeepSeek (*p* = 0.001). The same applies to the comparison of the ChatGPT 3.5 tool to Gemini Advanced 2.0 (*p* = 0.002) and Gemini Advanced 2.0 to DeepSeek (*p* = 0.01).

The results were confirmed by a statistically significant relationship between the level of substantive correctness of answers and the AI tool, χ^2^(6) = 27.92; *p* < 0.001. The highest level was observed for ChatGPT-4o, while the lowest concerned ChatGPT 3.5 and DeepSeek (Figure 7). In the *low* category, the distribution of responses across the tools was as follows: ChatGPT-4o—0%; ChatGPT-3.5—21%; DeepSeek—42%; Gemini Advanced 2.0—38%. For the *moderate* level, the proportions were 19%, 26%, 23%, and 32%, respectively, while for the *high* level, they were 35%, 25%, 25%, and 15%. Percentages presented in Figure 7 are normalized within each accuracy category (low, moderate, high), with the values across tools summing to 100% for each category.

To examine the statistically significant interaction between factors, a simple main effects analysis was conducted (Figure 8). It showed the following:-For the first analyzed question, the ChatGPT-4o tool obtained a higher mean of given answers compared to Gemini Advanced 2.0 (*p* = 0.02).-For the first analyzed question, the Gemini Advanced 2.0 tool obtained a lower mean of given answers compared to DeepSeek (*p* = 0.007).-For the third analyzed question, the ChatGPT-4o tool obtained a higher mean of given answers compared to Gemini Advanced 2.0 (*p* < 0.001).-For the third analyzed question, the Gemini Advanced 2.0 tool obtained a lower mean of given answers compared to ChatGPT-3.5 (*p* = 0.003) and DeepSeek (*p* < 0.001).-For the fourth analyzed question, the ChatGPT-4o tool obtained a higher mean of given answers compared to each of all the other tools (*p* < 0.001).-The ChatGPT-4o tool, for the second question, obtained a lower mean of correct answers compared to question 1 (*p* < 0.001), 3 (*p* = 0.001), and 4 (*p* < 0.001).-The ChatGPT-3.5 tool, for the second question, obtained a lower mean of correct answers compared to question 1 (*p* = 0.001) and 3 (*p* = 0.002). The same applies to the comparison of question 4 to 3 (*p* < 0.001).-The Gemini Advanced 2.0 tool obtained a higher mean of correct answers for question 1 compared to question 2 (*p* = 0.03) and 4 (*p* < 0.001).-The DeepSeek tool obtained a lower mean of correct answers for question 4 compared to each of the remaining questions (*p* < 0.001).

**Figure 8 jcm-14-07563-f008:**
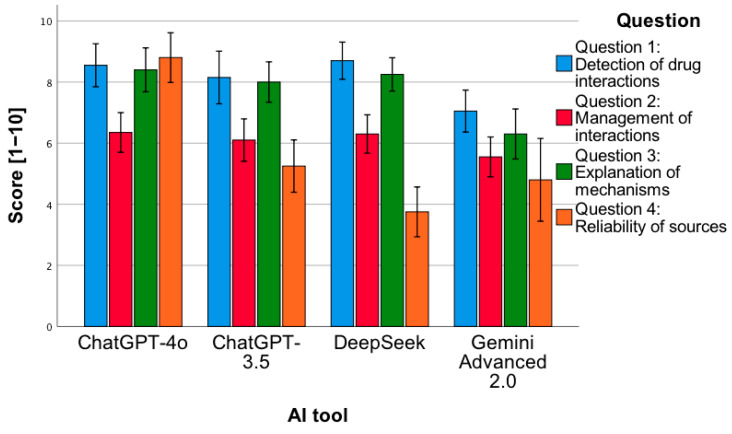
Correctness of answers provided by AI tools to 4 types of questions related to drug interactions.

### 3.3. Correctness of Answers from PESF—Hospital Pharmacy

In the next step, a general analysis of the correctness of the answers to 240 questions from PESF in the field of hospital pharmacy was conducted, namely 120 from 2018 and the same number from 2019. No statistically significant relationship was observed, χ^2^(3) = 1.55; *p* = 0.67. The percentage of correct answers ranged from 83% to 86% (Figure 9).

## 4. Discussion

This study represents the first systematic comparison of AI tools’ responses with actual clinical cases described in the scientific literature, including the pharmacotherapy used. Analyzing the responses provided by individual AI tools to questions regarding the correctness of pharmacotherapy and drug interactions, it can generally be observed that the ChatGPT-4o, ChatGPT-3.5, and DeepSeek models achieved a higher level of accuracy in their answers compared to Gemini Advanced 2.0. The responses generated by Gemini Advanced 2.0 were usually lengthy and imprecise and often lacked the unambiguous information expected in the context of the question. The lack of structure and highlighting of key content made them difficult to read. In many cases, this model avoided giving specific answers, citing ethical considerations such as the risk of following treatment without consulting a physician.

Our results are consistent with the findings of a study in which the ChatGPT, Copilot, and Gemini models were compared in terms of their responses to clinical questions. In both studies, the Gemini model was characterized by lower precision and limited substantive accuracy of the generated content. The ChatGPT models achieved higher results, confirming their greater potential in educational and clinical applications. Gemini did not meet the criteria enabling its use as a tool to support clinical decision-making [55]. In the present study, the impact of prompting techniques on the content and quality of recommendations, as well as the usefulness of the presented information sources, was also evaluated, allowing these findings to be related to the earlier results. The study results did not show statistically significant differences between the responses provided by AI tools with or without the use of prompts. This relationship was independent of the type of context and the form of the query used. Although this is the first study of its kind in the field of pharmacotherapy, it should be noted that similar analyses have already been conducted in the broader context of clinical medicine. Their results, however, differ and suggest that the use of prompts significantly improves the accuracy of responses. An example is a study that evaluated the effectiveness of various prompt engineering techniques in clinical Natural Language Processing (NLP) tasks using the GPT-3.5, Gemini, and Large Language Model Meta AI 2 (LLaMA-2) models. The analysis showed that tailoring prompts to a specific task significantly increased response accuracy, with heuristic and chain of thought prompts achieving the highest effectiveness. It was also shown that few-shot prompting improved results in more complex scenarios, while ensemble approaches made it possible to combine the strengths of different types of prompts. Consequently, GPT-3.5 consistently outperformed the other models in terms of response quality in most of the analyzed tasks [53]. Based on another study, it was shown that the effectiveness of language models in terms of compliance with clinical guidelines depends on the type of prompt used. The analysis conducted with reference to the recommendations of the American Academy of Orthopaedic Surgeons regarding the treatment of osteoarthritis found that GPT-4-Web with ROT prompting achieved the highest response consistency. However, it was emphasized that the reliability of individual models and prompts was not stable, which indicates significant limitations in the predictability of the obtained results [54]. The differences between our results and those of the cited studies may stem from the distinct clinical context and task design. The earlier works [53,54] focused primarily on general clinical NLP or guideline adherence tasks, where structured prompts directly influenced text interpretation and evidence extraction. In contrast, the present analysis evaluated pharmacotherapy decision-making based on complex case reports, where reasoning depends not only on the structure of the prompt but also on the model’s ability to synthesize heterogeneous clinical data. Moreover, unlike zero- or few-shot NLP experiments, the present study used open-ended, context-driven prompts that more closely mimic real clinical queries. This methodological distinction likely explains the lack of a statistically significant difference between prompted and non-prompted responses observed in our results. These differences may result both from different methodological assumptions and from potential errors in the subjective evaluation of clinical cases. Another analyzed aspect was drug interactions. To our knowledge, the conducted study represents the first work in which the responses of artificial intelligence tools were evaluated in relation to drug interactions described in actual patients in the scientific literature. The literature highlights the growing role of machine learning and deep learning in identifying connections between drugs and their molecular targets. Directions for further work are also indicated, including improving the quality of input data, increasing model transparency, and verifying models’ usefulness in clinical practice [56]. Studies published to date have evaluated the effectiveness of ChatGPT in predicting and explaining drug interactions based on 40 drug pairs selected from the scientific literature. The analysis conducted using a two-step question scheme showed that the vast majority of responses were correct, although they were often inconclusive. Additionally, it was found that the generated content was characterized by low readability and required a high level of education for full understanding, which limits its practical application by patients [57]. In one study, the effectiveness of ChatGPT-3.5, ChatGPT-4, Bing AI, and Bard in predicting drug interactions based on 255 drug pairs was compared. The highest sensitivity and specificity were recorded for Bing AI, which also achieved the greatest accuracy, while the weakest results were obtained by ChatGPT-3.5. The analysis also revealed high variability in accuracy and specificity depending on the class of drugs, particularly in the case of ChatGPT-3.5, ChatGPT-4, and Bard [58]. In another study, the usefulness of ChatGPT-3.5 in predicting drug interactions in clinical settings was assessed using data from 120 hospitalized patients. The results indicated low sensitivity and minimal agreement with the pharmacists’ assessment, confirming the limited clinical utility of this tool in clinical practice [59].

For hospital pharmacy exam questions, the models scored 83–86%, indicating potential for solving closed-format questions. A similar issue was addressed in a study that evaluated the effectiveness of GPT models in solving questions from the Korean Pharmacist Licensing Examination. It was shown that GPT-4 achieved an average of 86.5% correct answers, surpassing GPT-3.5, which scored 60.7%. The best results were observed in the area of biopharmacy, while the lowest effectiveness was recorded in health legislation [60]. Similar analyses were also conducted on the pharmacist licensing exam in Taiwan. In this study, the effectiveness of ChatGPT-3.5 in answering questions in Chinese and English was evaluated, with the average percentage of correct answers ranging from 54.4 to 56.9% in the first stage and 53.8 to 67.6% in the second. In the Chinese-language test, only pharmacology and pharmaceutical chemistry reached the passing threshold, while the results in English were significantly higher in most sections, especially in clinical pharmacy and pharmacotherapy. Ultimately, the model did not pass the exam, which indicates its limited usefulness but at the same time highlights the potential for further improvement and applications in pharmaceutical education [61]. In the pharmacist licensing exam in Japan, the GPT-4 model achieved 72.5% correct answers, which allowed it to meet the passing criterion. For questions without graphic elements, effectiveness reached 80%, clearly surpassing the results of GPT-3.5, while questions containing diagrams remained a significant limitation [62].

In summary, the conducted analyses indicate that AI tools demonstrate that effectiveness varies by domain and task. They are capable of generating substantively correct answers, but they often lack precision, consistency, and sufficient readability. Comparisons with the literature show that the results obtained in different clinical and educational contexts are not uniform and depend on the adopted methodology. Prompting techniques do not always lead to the expected improvement in response quality. This points to the need for further refinement of the models and a more in-depth evaluation of their potential in clinical and educational practice.

## 5. Limitations

This study has several important limitations that should be considered when interpreting the results. First, the analysis was based on clinical cases described in the scientific literature, which limits the ability to fully reflect the complexity of real clinical situations. Such an approach may lead to the overrepresentation of certain therapeutic scenarios and fail to account for the diversity of clinical practice. Second, the evaluation of responses generated by the models was partly subjective. However, it was carried out by the authors with experience in clinical pharmacy, which minimized the risk of misinterpretation. The study involved a limited number of AI tools. Given rapid model development, our conclusions apply only to the analyzed model versions. Another limitation concerns the knowledge cutoff dates of the tested models, which preceded some of the most recent 2024–2025 case reports; therefore, the models may not have had access to the newest literature, and their responses were based solely on the case descriptions provided. An additional limitation is the lack of assessment of response reproducibility across different sessions, which could reveal variability resulting from the nature of generative models. Finally, the study did not include aspects of the practical implementation of AI tools in the clinical environment, such as integration with information systems or patient safety, which limits the possibility of directly translating the obtained results into practice.

## 6. Conclusions

AI tools can support the analysis of pharmacotherapy issues and drug interactions; however, their effectiveness varies and depends on the model used. The obtained results indicate the need for further improvement of these solutions and caution in their use in clinical and educational practice.

## Figures and Tables

**Figure 1 jcm-14-07563-f001:**
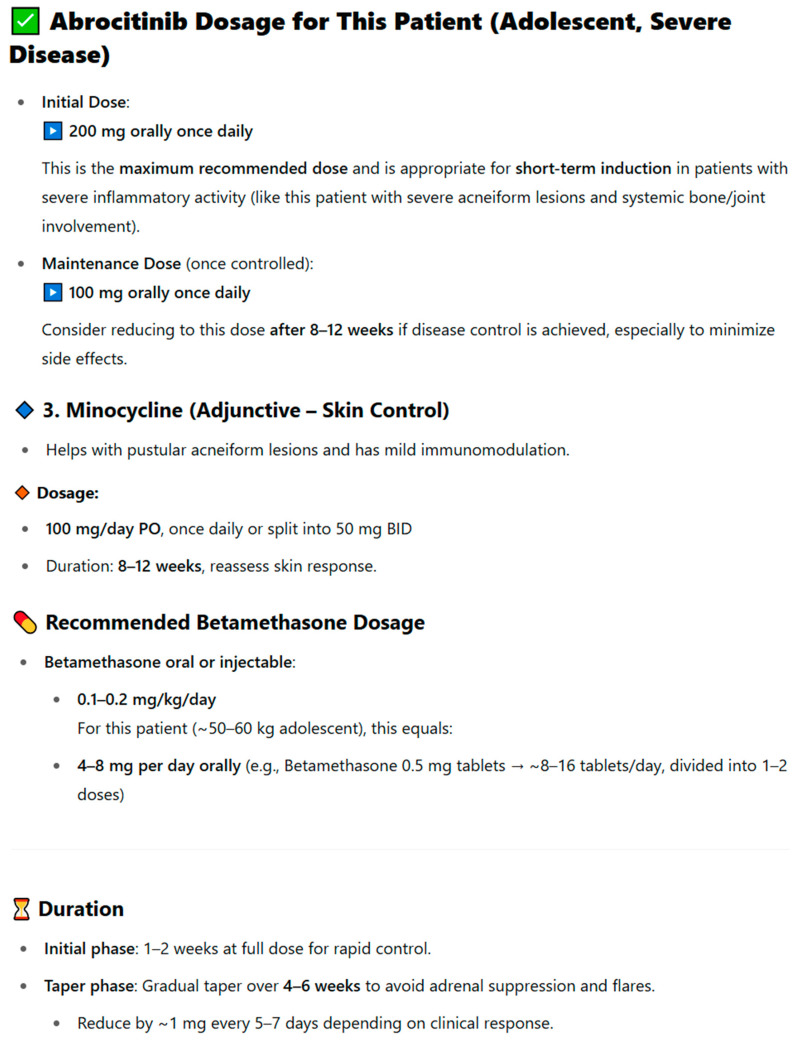
Example of the ChatGPT 3.5 model’s response to question 3 concerning the dosing of pharmacotherapy in an individual clinical case.

**Figure 2 jcm-14-07563-f002:**
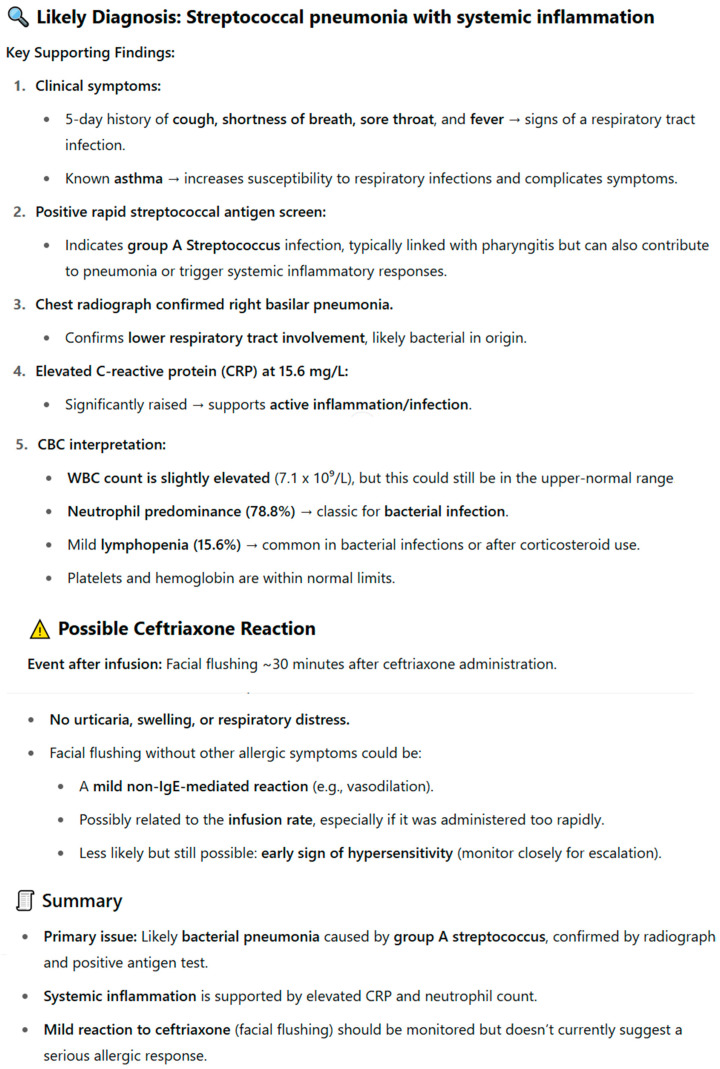
Response of the ChatGPT 3.5 model concerning the analysis of drug interactions in the case of a disulfiram-like reaction between prednisolone and ceftriaxone.

**Figure 3 jcm-14-07563-f003:**
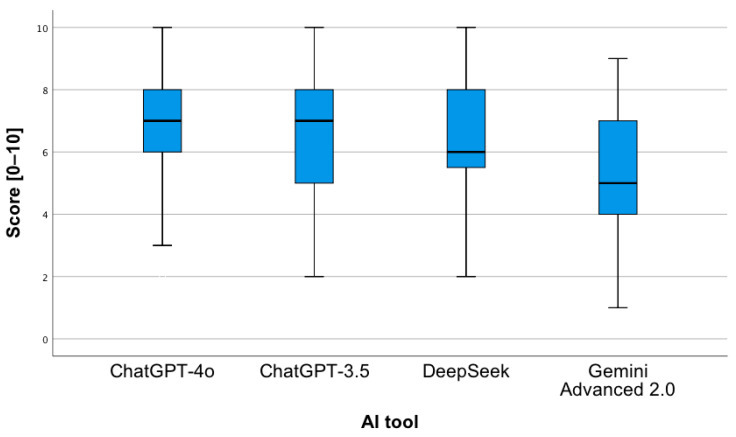
Correctness of answers provided by AI tools to questions concerning the proposed pharmacotherapy.

**Figure 4 jcm-14-07563-f004:**
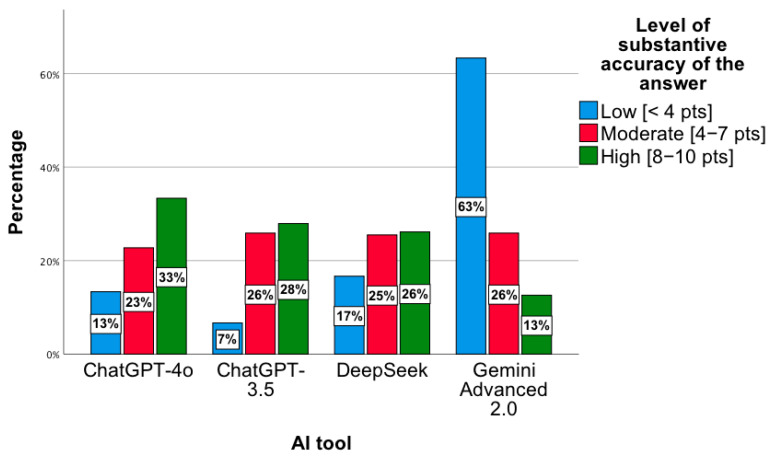
Level of substantive correctness of answers to questions concerning the proposed pharmacotherapy by individual AI tools.

**Figure 5 jcm-14-07563-f005:**
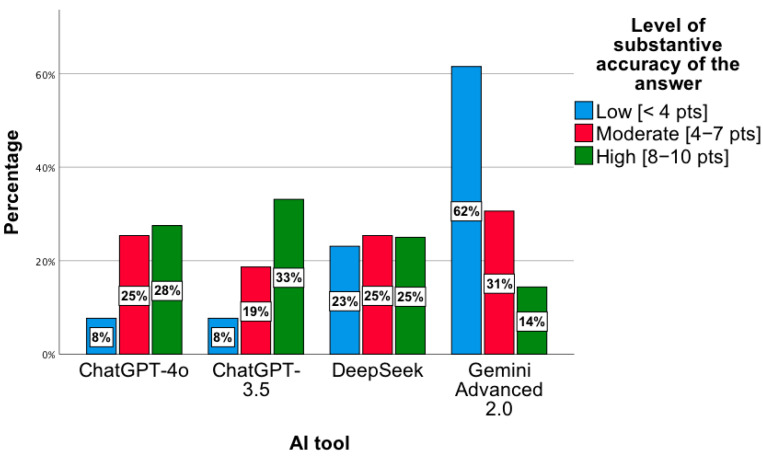
Level of substantive correctness of answers to precise questions concerning the proposed pharmacotherapy by individual AI tools.

**Figure 6 jcm-14-07563-f006:**
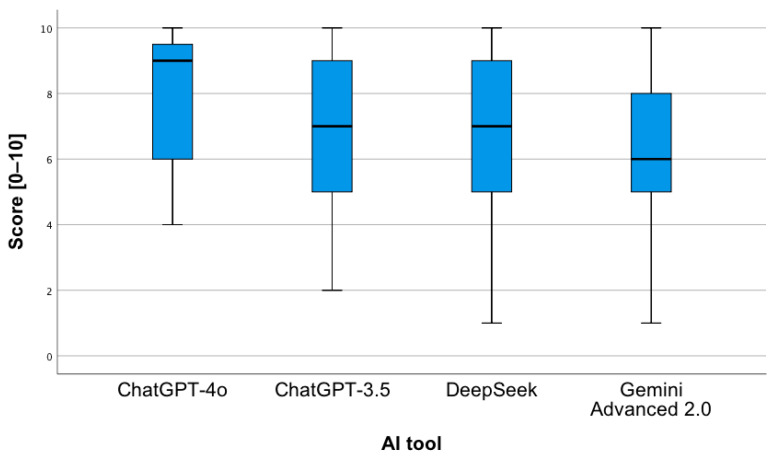
Correctness of answers provided by AI tools to questions concerning drug interactions.

**Figure 7 jcm-14-07563-f007:**
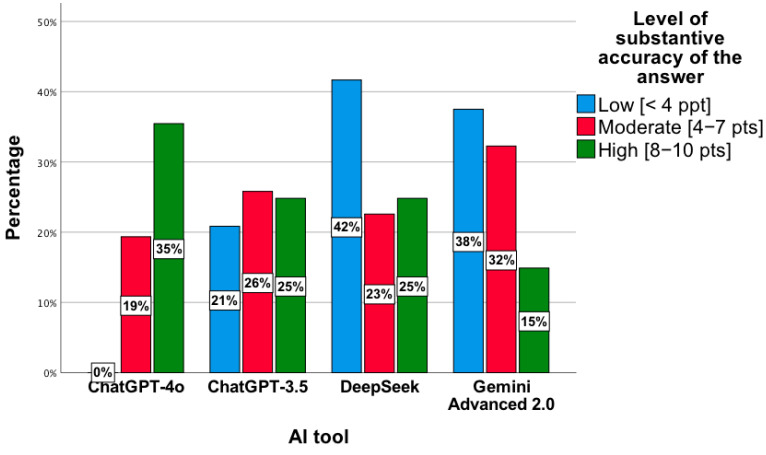
Level of substantive correctness of answers to questions concerning drug interactions by individual AI tools.

**Figure 9 jcm-14-07563-f009:**
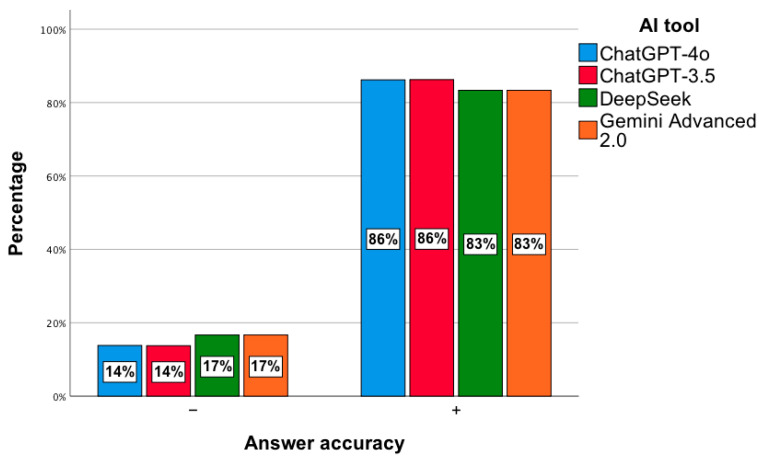
Correctness of answers provided by AI tools to exam questions from PESF—hospital pharmacy.

**Table 1 jcm-14-07563-t001:** Descriptive statistics concerning the correctness of answers provided by AI tools to the analyzed precise questions regarding the proposed pharmacotherapy.

Narzędzie AI	M	SD	Min	Max	Q1	Me	Q3
ChatGPT-4o	7.13	1.75	2	10	6	7	9
ChatGPT 3.5	7.3	1.67	3	10	6	7	8.75
Gemini Advanced 2.0	5.33	1.97	1	9	4	5	7
DeepSeek	6.49	2.04	2	10	5	6.5	8

Abbreviations: M, mean; SD, standard deviation; Min, minimum; Max, maximum; Q1, first quartile; Me, median; Q3, third quartile.

## Data Availability

Data supporting the reported results are available from the corresponding author upon reasonable request.

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
