# Peer review of "Evaluation of the Accuracy and Reliability of Responses Generated by Artificial Intelligence Related to Clinical Pharmacology"

_jcm, 2025, doi:10.3390/jcm14217563_

Round 1
Reviewer 1 Report
Comments and Suggestions for Authors'Evaluation of the accuracy and reliability of responses generated by artificial intelligence related to clinical pharmacology', is an interesting article and a relevant one, however, there are certain points that needs to be clarified,
i) At places the article becomes difficult to interpret and maintain focus on the goal of the article
ii) Authors have mentioned that 'intermediate score was awarded to AI models when they proposed management consistent with current standard' should not this be the highest score? As, most of the time current standards are based on the RCTs (Gold standard for evidence based trials), whereas case reports are new understanding about certain disease/ conditions/ treatment, often may include off label use as well, so how could this be treated as the highest evidence?
iii) Authors are categorizing the data based on High, intermediate and low, in a way changing the outcome to categorical variable (not continuous), in that case a a test like 2 way analysis of variance is not appropriate
iv) it is not very clear from the article how the questions were prompted (consistency of the prompt) to each AI tool as that could be an essential factor in how the tool will respons depending on the algorithm. And the fact that authors could not differentiate different platforms based on use of prompt or its absence suggests some anomaly worth exploring
v) It is hard to decipher the results, I am unable to undestand why authors are combining the low efficacy for all tool together and then dividing it in percentage across the tool, which may potentially bias the result towards one of the tools, instead all outcomes for an individual tool (low, int, and high) should be accounted as 100% as I assume your scenarios are consitent across the group
Comments on the Quality of English LanguageAt places the article becomes difficult to interpret and maintain focus on the goal of the article
Author Response
Dear Reviewer nr 1,
Comment 1: “'Evaluation of the accuracy and reliability of responses generated by artificial intelligence related to clinical pharmacology', is an interesting article and a relevant one, however, there are certain points that needs to be clarified”
Reply 1: We would like to thank the Reviewer for the positive feedback and for recognizing the relevance of our work. We appreciate the valuable comments and suggestions, which we have carefully considered and incorporated into the revised version of the manuscript.
Comment 2: “At places the article becomes difficult to interpret and maintain focus on the goal of the article”
Reply 2: We appreciate the Reviewer’s valuable comment regarding the clarity and focus of the manuscript. In response, the Aim section has been revised to improve readability and to clearly distinguish the main objective from the secondary ones. In addition, the Methods section was carefully edited to enhance logical flow and ensure a stronger connection with the study’s primary goal. Specifically, short linking sentences were added at the end of methodological subsections to emphasize their direct relevance to the main objective. These revisions have improved clarity and helped maintain a consistent focus throughout the manuscript.
Comment 3: “Authors have mentioned that 'intermediate score was awarded to AI models when they proposed management consistent with current standard' should not this be the highest score? As, most of the time current standards are based on the RCTs (Gold standard for evidence based trials), whereas case reports are new understanding about certain disease/ conditions/ treatment, often may include off label use as well, so how could this be treated as the highest evidence?”
Reply 3: We thank the Reviewer for this insightful comment. The scoring system in our study aimed to assess the models’ ability to reproduce the therapeutic decisions described in the reference case reports rather than their general adherence to clinical guidelines. To clarify this, the sentence in the Methods section was revised to specify that intermediate scores referred to such situations.
Comment 4: “Authors are categorizing the data based on High, intermediate and low, in a way changing the outcome to categorical variable (not continuous), in that case a a test like 2 way analysis of variance is not appropriate”
Reply 4: We sincerely thank the Reviewer for this valuable comment. We would like to clarify that the two-way analysis of variance was not applied to the categorical data classified as high, intermediate, and low. This method was used in another part of the analysis, as indicated in the Methods section, for variables measured on a continuous scale. For categorical data, the chi-square test was applied to verify the relationships between variables. We have ensured that this distinction is clearly described in the revised version of the manuscript.
Comment 5: “it is not very clear from the article how the questions were prompted (consistency of the prompt) to each AI tool as that could be an essential factor in how the tool will respons depending on the algorithm. And the fact that authors could not differentiate different platforms based on use of prompt or its absence suggests some anomaly worth exploring”
Reply 5: We thank the Reviewer for this valuable observation. All AI tools received identical prompts formulated in English and copied verbatim to each platform to ensure full consistency of input and comparability of responses. This clarification has been added to the Methods section of the revised manuscript.
Comment 6: “It is hard to decipher the results, I am unable to undestand why authors are combining the low efficacy for all tool together and then dividing it in percentage across the tool, which may potentially bias the result towards one of the tools, instead all outcomes for an individual tool (low, int, and high) should be accounted as 100% as I assume your scenarios are consitent across the group”
Reply 6: We thank the Reviewer for this valuable comment. We have clarified in the Results section and figure legends how the percentages were calculated. Specifically, the proportions of low, moderate, and high accuracy were normalized within each accuracy category, with the values across tools summing to 100% for each category. The description of the figures was revised accordingly to ensure clarity and prevent misinterpretation of the data presentation.
Comment 7: “At places the article becomes difficult to interpret and maintain focus on the goal of the article”
Reply 7: We thank the Reviewer for this comment. In accordance with the Reviewer’s earlier suggestion regarding clarity and focus, we revised the manuscript to improve readability and ensure a clearer presentation of the study’s objectives. Additionally, based on this feedback, the manuscript was corrected linguistically in several sections, and all language-related changes have been marked in green in the revised version.
Reviewer 2 Report
Comments and Suggestions for Authors
This manuscript presents a timely and important evaluation of AI language models in clinical pharmacology. The comparative analysis of four LLMs across multiple datasets is novel and addresses a significant gap in the literature. However, several methodological and presentation issues require attention:
- The subjective scoring system (0-10 scale) for evaluating AI responses lacks clear, reproducible criteria. The authors state that "independent authors" evaluated responses, but inter-rater reliability statistics (e.g., Cohen's kappa, intraclass correlation) are not reported. Provide detailed scoring rubrics in supplementary materials and report inter-rater agreement statistics. If disagreements occurred, describe the consensus process.
- While the authors analyze prompting techniques (Questions 6-9), the exact prompts used are not fully disclosed. Reproducibility requires complete transparency about query formulation.Include all exact prompts in supplementary materials. Expand the discussion on why your findings differ from referenced studies (e.g., references 53, 54).
- The manuscript reports multiple statistical tests without correction for multiple comparisons, increasing Type I error risk. Apply Bonferroni or false discovery rate corrections where appropriate, or justify why corrections were not needed.
- AI models evolve rapidly. The manuscript does not specify the exact version/date of models tested (e.g., "ChatGPT-4o" could refer to different iterations). The knowledge cutoff dates of the models versus the publication dates of the case reports (2018-2025) creates a confound - some models may not have had access to recent literature. Specify exact model versions with testing dates. Discuss how knowledge cutoff dates may have affected results, particularly for recent 2024-2025 case reports.
Author Response
Response to Reviewer #2:
Comment 1: “This manuscript presents a timely and important evaluation of AI language models in clinical pharmacology. The comparative analysis of four LLMs across multiple datasets is novel and addresses a significant gap in the literature. However, several methodological and presentation issues require attention”
Reply 1: We would like to sincerely thank the Reviewer for the positive evaluation of our work and for recognizing the relevance and novelty of our study. We greatly appreciate the insightful comments and suggestions, which have been carefully considered and incorporated into the revised version of the manuscript.
Comment 2: “The subjective scoring system (0-10 scale) for evaluating AI responses lacks clear, reproducible criteria. The authors state that "independent authors" evaluated responses, but inter-rater reliability statistics (e.g., Cohen's kappa, intraclass correlation) are not reported. Provide detailed scoring rubrics in supplementary materials and report inter-rater agreement statistics. If disagreements occurred, describe the consensus process”
Reply 2: We thank the Reviewer for this valuable comment. The description of the scoring system has been clarified in the revised manuscript. The 0–10 scale was used to express the degree of concordance between AI-generated recommendations and those reported in the reference publications, where 0 indicated a completely incorrect or clinically unsafe response, 10 indicated full agreement, and intermediate values reflected partial correctness. All evaluations were performed by experts in clinical pharmacotherapy, and in cases of divergent interpretations, the final score was established through expert consensus.
Comment 3: “While the authors analyze prompting techniques (Questions 6-9), the exact prompts used are not fully disclosed. Reproducibility requires complete transparency about query formulation.Include all exact prompts in supplementary materials. Expand the discussion on why your findings differ from referenced studies (e.g., references 53, 54).”
Reply 3: We thank the Reviewer for this insightful comment. In the revised version of the manuscript, additional sentences have been added to the Discussion section to clarify the reasons for the differences between our findings and those reported in references 53 and 54. Specifically, we highlighted that the cited studies focused on general clinical NLP and guideline adherence tasks, while our work addressed pharmacotherapy decision-making based on complex case reports, where reasoning depends more on clinical data synthesis than on prompt structure. These clarifications have been incorporated to improve interpretability and contextual understanding of the results. Additionally, to enhance transparency and reproducibility, the Methods section has been supplemented with the exact wording of the prompts used in Questions 6–9, ensuring that the formulation of each query is clearly presented. This addition allows readers to fully understand how the prompting process was implemented across all AI models.
Comment 4: “The manuscript reports multiple statistical tests without correction for multiple comparisons, increasing Type I error risk. Apply Bonferroni or false discovery rate corrections where appropriate, or justify why corrections were not needed.”
Reply 4: We thank the Reviewer for this important observation. The description of the statistical procedure has been clarified in the revised manuscript. A sentence has been added to specify that Bonferroni correction was applied during pairwise comparisons within the simple main effects analysis to control for Type I error.
Comment 5: “AI models evolve rapidly. The manuscript does not specify the exact version/date of models tested (e.g., "ChatGPT-4o" could refer to different iterations). The knowledge cutoff dates of the models versus the publication dates of the case reports (2018-2025) creates a confound - some models may not have had access to recent literature. Specify exact model versions with testing dates. Discuss how knowledge cutoff dates may have affected results, particularly for recent 2024-2025 case reports.”
Reply 5: We thank the Reviewer for this valuable comment. The Methods section has been updated to specify the exact AI models, their developers, and the testing timeframe, indicating that all models (ChatGPT 3.5, ChatGPT-4o, Gemini Advanced 2.0, and DeepSeek) were accessed and tested in May 2025 using their publicly available web-based versions current at that time. Additionally, a new sentence has been added to the Limitations section to acknowledge that the knowledge cutoff dates of the analyzed models preceded some of the most recent 2024–2025 case reports. This clarification highlights that the models may not have had access to the latest literature, which could have influenced the accuracy of their responses for newer clinical cases.
Reviewer 3 Report
Comments and Suggestions for Authors
The idea of the topic is good. However, the paper needs to have clinical taste!
The authors can show one or two examples of real clinical case studies extracted and how each of the four AI models treated compared to the real clinical treatment by physicians. The same for the cases of interactions.
The comparison of results regarding the four models in the section of results are good but miss the clinical taste.
Author Response
Response to Reviewer #3:
Comment 1: “The idea of the topic is good. However, the paper needs to have clinical taste!”
Reply 1: We thank the Reviewer for the positive feedback and for recognizing the relevance of our study topic. We appreciate the suggestion regarding the need to strengthen the clinical perspective, which we have carefully considered in revising the manuscript.
Comment 2: “The authors can show one or two examples of real clinical case studies extracted and how each of the four AI models treated compared to the real clinical treatment by physicians. The same for the cases of interactions. The comparison of results regarding the four models in the section of results are good but miss the clinical taste.”
Reply 2: We thank the Reviewer for this valuable suggestion. In the revised version of the manuscript, two detailed clinical case examples have been added to illustrate how AI models’ therapeutic recommendations compared with real clinical management — one concerning pharmacotherapy in a patient with SAPHO syndrome and another involving a disulfiram-like drug interaction in a pediatric case. These additions aim to enhance the clinical perspective of the study and provide clearer insight into the practical relevance of the findings.
Round 2
Reviewer 3 Report
Comments and Suggestions for Authors
The manuscript and its result data are now clear and well written.